# Lipoprotein(a): Cardiovascular Disease, Aortic Stenosis and New Therapeutic Option

**DOI:** 10.3390/ijms24010170

**Published:** 2022-12-22

**Authors:** Alessandro Maloberti, Saverio Fabbri, Valentina Colombo, Elena Gualini, Massimiliano Monticelli, Francesca Daus, Andrea Busti, Michele Galasso, Lorenzo De Censi, Michela Algeri, Piera Angelica Merlini, Cristina Giannattasio

**Affiliations:** 1Cardiology 4, Cardio Center A. De Gasperis, ASST GOM Niguarda, 20162 Milan, Italy; 2School of Medicine and Surgery, Milano-Bicocca University, 20126 Milan, Italy

**Keywords:** lipoprotein(a), cardiovascular events, aortic stenosis

## Abstract

Atherosclerosis is a chronic and progressive inflammatory process beginning early in life with late clinical manifestation. This slow pathological trend underlines the importance to early identify high-risk patients and to treat intensively risk factors to prevent the onset and/or the progression of atherosclerotic lesions. In addition to the common Cardiovascular (CV) risk factors, new markers able to increase the risk of CV disease have been identified. Among them, high levels of Lipoprotein(a)—Lp(a)—lead to very high risk of future CV diseases; this relationship has been well demonstrated in epidemiological, mendelian randomization and genome-wide association studies as well as in meta-analyses. Recently, new aspects have been identified, such as its association with aortic stenosis. Although till recent years it has been considered an unmodifiable risk factor, specific drugs have been developed with a strong efficacy in reducing the circulating levels of Lp(a) and their capacity to reduce subsequent CV events is under testing in ongoing trials. In this paper we will review all these aspects: from the synthesis, clearance and measurement of Lp(a), through the findings that examine its association with CV diseases and aortic stenosis to the new therapeutic options that will be available in the next years.

## 1. Introduction

Cardiovascular (CV) diseases are the first cause of death worldwide, accounting for 17.8 million deaths/year [1,2,3]. Atherosclerosis is a chronic and progressive inflammatory process which begins early in life (during the second decade) and has late clinical manifestation. This slow pathological trend shows the importance to early identify high-risk patients and the importance of intensive treatment of identified risk factors to prevent the onset and/or the progression of atherosclerotic lesions. In addition to the common CV risk factors (age, sex, genetic heritage, dyslipidemia, diabetes mellitus, arterial hypertension, smoking, obesity and sedentary lifestyle) other new markers able to increase the risk of CV diseases have been identified. Among them, high levels of lipoprotein(a)—Lp(a) lead to a very high risk of future CV diseases [4,5,6,7,8,9,10,11,12] and new aspects have been recently identified, such as its association with aortic stenosis. Although untill recent years it has been considered an unmodifiable risk factor, specific drugs have been developed with a strong efficacy in reducing the circulating levels of Lp(a); their capacity to reduce subsequent CV events is under testing in ongoing trials. Lp(a) is mainly a monogenic cardiovascular risk factor (70–90% of its circulating levels are genetically determined) and its association with future CV diseases remains significant also when dyslipidemia is well-controlled (i.e., Low-Density Lipoprotein—LDL—under the 55 mg/dL target) [13,14].

## 2. Lipoprotein(a) Molecule: Structure and Determinants of Plasma Levels

Lp(a) is an atherogenic lipoprotein consisting of a polymorphic glycoprotein apo(a) covalently linked to an apoB100-containing lipoprotein [15]. It was first discovered by the Norwegian physician Kare Berg in 1963 and was initially considered an antigenic variant of LDL cholesterol due to the presence of an antigen called Lp(a). In the 1970s, a Swedish researcher identified a new electrophoretic lipoprotein variant associated with CV disease, initially called Lp prebeta-1 and later identified as being the Lp(a) [16].

The efforts of several research groups resulted in the isolation and purification of Lp(a) [17], which provided key insights into the structural and biochemical characteristics of this lipoprotein. In the late 1980s, the sequencing of a cDNA corresponding to the gene encoding apo(a) demonstrated that it had evolved through duplication of the plasminogen gene with which it shares a structural homology. Plasminogen has a five triloop structure called Kringles (K, progressively numbered as KI, KII, KIII, KIV and KV) and a protease domain. Instead, apo(a) has only one copy of KV, one inactive serine protease-like domain, and ten subtypes of KIV (KIV1 a KIV10) due to different amino-acid replacements. KIV type 1 (KIV1) and types 3–10 (KIV3–10) are present as a single copy in all apo(a) species, whereas KIV2 is present in a variable number of identical repeats [18] (Figure 1).

Specific properties relevant for the assembly and molecular pathology of Lp(a) are attributable to Lysine-Binding Sites (LBS), that could be present in some of the KIV domains of apo(a). Weak LSB present in KIV7 and 8 are required for non-covalent interactions with specific lysine domains in the amino-terminal globular domain of apoB-100. This non-covalent association is the first step of Lp(a) assembly and is followed by the formation of a covalent disulfide linkage between apoB-100 cysteine 4326 and the unique unpaired cysteine residue (Cys4057) in apo(a) KIV9. A further important domain is in the KIV10 where a strong LBS mediates interactions with fibrin, but also it contains the site where a pro-inflammatory oxidized phospholipid (generated by polyunsaturated fatty acid oxidation) is covalently attached [19]. These domains are important for the pro-inflammatory and pro-thrombotic properties of Lp(a).

Lp(a) synthesis is genetically regulated with minimal or no influence from dietary and environmental factors [20]. Genetic is the main determinant of plasma levels and it depends on the codominant expression of 2 apo(a) alleles (located at positions 26 and 27 on the long arm of chromosome 6—6q26-27). Final circulating Lp(a) levels are determined by the sum contributed by each allele. Most of the people have 2 detectable circulating Lp(a) isoforms, each arising from apo(a) of different sizes; the smallest is usually the most represented [21,22]. Alleles encode as few as 1 and as many as 34 KIV2 repeats from 1 to 34 KIV2 repeats, giving final apo(a) isoforms containing between 10 and 43 KIV-like domains (Figure 1).

However, Lp(a) concentration does not only depend on gene alleles and apo(a) size, as shown in genetically unrelated individuals that have the same isoform size combination where plasma concentration can vary by up to 200-fold [15]. The first of these further factors is that numerous Single Nucleotide Polymorphisms (SNPs) in the apo(a) locus strongly associate with Lp(a) levels [21,22]. In a recent Genome-Wide Association Studies (GWAS) meta-analysis, in 13,781 individuals many different SNPs have been identified in a 1.76 MB large region around the apo(a) gene [23,24,25]. Furthermore, 48 of them were independently associated with Lp(a) concentrations [25]. A study by Morgan et al. investigated two very rare variants, R990Q (rs41259144) located in the KIV type 4 and R1771C (rs139145675) located in the KV, which were in four null Lp(a) individuals. These two variants probably impair the protein folding and so its secretion process [26].

## 3. Lipoprotein(a) Molecule: Synthesis and Clearance

Lp(a) is synthesized in the hepatocyte [27,28] and secreted into the plasma. Its circulating levels are predominantly determined by the production rather than by its clearance [29,30,31].

The assembly of Lp(a) from apo(a) and apoB and the pathways for Lp(a) removal from the circulation are not completely defined [19]. Apo(a) synthesis and secretion follow different steps: transcription of the apo(a) gene, translation (identified as the crucial step in secretion rate), post-translational modifications needed for the folding of apo(a), movement through the secretory pathway and assembly of Lp(a) particles [16]. Several studies have identified transcription factor-binding sites in the apo(a) gene promoter region that can regulate gene expression. These include: (1) an interleukin-6 responsive element in the promoter that induces transcription; (2) a Direct Repeat 1- promoter element that binds bile acid-liganded farnesoid X receptor and decreases transcription; (3) an E26 transformation-specific motif that binds erythroblast transformation specific transcription factor ELK1 to mediate fibroblast growth factor-19 transcriptional repression; (4) and multiple cyclic adenosine monophosphate responsive elements thought to mediate repression of apo(a) transcription by niacin [32].

Despite being folded at the same rate as smaller isoforms, large apo(a) isoforms are retained longer in the endoplasmic reticulum and are subjected to increased degradation by the proteasome. This mechanism contributes to the general inverse correlation between apo(a) isoform size and plasma Lp(a) levels [18,19]. 

The site of assembly is not well defined: according to in vitro studies the binding between apo(a) and apoB may occur either at the surface of the hepatocytes or in the space of Disse [21]. In vivo studies with stable isotopes show an intracellular assembly instead [16].

Even though the clearance of Lp(a) is mainly determined by the liver [29], the presence of an arteriovenous difference of its concentrations in the renal circulation [30] and of apo(a) fragments in the urine [33] has also suggested that kidneys intervene in the process. Indeed, people with chronic kidney disease also present high values of Lp(a) [34] with the excretion reduction that begins at a glomerular filtration rate of 70 mL/min/mq.

Lp(a) can interact with different receptors classes: LDL receptors (LDLR), scavenger receptors, toll like receptors, carbohydrate receptors (lectins), and plasminogen receptors [35]. Among them, due to the similarity between Lp(a) and LDL, the LDLR is the most involved in its clearance. Kinetic studies demonstrated a longer circulating time for Lp(a) than LDL, probably due to the smaller affinity for LDLR than LDL molecules related to apo(a) particle interfering in receptor binding [21].

However, the approbative trials of proprotein convertase subtilisin/kexin type 9 (PCSK9) inhibitors describe a Lp(a) reduction in patients treated with PCSK9 inhibitors, probably related to the reduction of degradation of all members of the LDLR family by PCSK9 protein and LDLR becoming more effective in a setting of low LDL-C [16].

Among the LDLR, very low density lipoprotein receptors (VLDLR), LDL receptor-related protein 1 (LRP1) and LDL receptor related protein 2 (LRP2), also called “megaline”, are involved in Lp(a) uptake [35]. LRP1 is highly expressed in the liver, playing an important role in chylomicron-remnant-metabolism [22]. LRP2 is instead highly expressed in kidneys and it is probably involved in the renal clearance of Lp(a) [36]. VLDLR did not show a significant interaction with apo(a) in the liver so it does not seem to have an important role in Lp(a) clearance [35].

## 4. Lipoprotein(a) Measurements and Cut-Off

The unique Lp(a) features, such as high heterogeneity, the covalent bond between apo(a) and apoB and its homology with plasminogen, have long been a major challenge in the development of a suitable and reliable immunoassay for its measurement. The first highly sensitive assay for Lp(a) measurements was reported in the 1970s by Albers et al. [37]. This historic approach assumed that the mass of the Lp(a) components was constant, while the well-known extreme size variability of apo(a) shows that it was a mistake. Expressing Lp(a) as a mass concentration (milligrams/deciliters; mg/dL) introduces an inherent bias because a given mass of Lp(a) represents a lesser number of particles for large isoforms and a greater number of particles for small isoforms. Moreover, the conversion of mass concentrations to particle concentrations using a single conversion factor of 2.5 leads to an overestimation of the concentration of large isoforms and an underestimation of the concentration of small isoforms.

Various immunochemical methods, such as ELISA, immunoturbidimetry, nephelometry and dissociation enhanced lanthanide immunoassay were employed in Lp(a) evaluation in serum or plasma using Monoclonal Antibodies (mAb) versus apo(a). Immunoassays are based on the measurement of signals due to antigen–antibody interaction. Two categories of immunoassays are involved in Lp(a) measurement. The “isoform dependent” method evaluates the entire protein mass, reported in mg/dL, including lipids, proteins and carbohydrates, and it is strongly related to the number of KIV2 [38]. Extreme apo(a) isoform size variability is associated with an overestimation of Lp(a) levels in patients with large apo(a) isoforms, and an underestimation in those with small apo(a) isoforms [21,39], leading to incorrect CV risk assessment. The “isoform-independent” method reports Lp(a) in nmol/L. relies on a direct binding of a double mAb, in which mAb a-6 is directed to an epitope present in apo(a) KIV2, while the detection antibody (a-40) is directed to an epitope present in apo(a) KIV9 [40], the unique nonrepeating KIV subtype. The latter is internationally considered the golden methodology standard because it is not influenced by different apo(a) sizes [16].

Recently, a mass spectrometry method has been developed and validated for Lp(a) measurement standardization. It solves the problem of size polymorphisms by selecting specific quantification peptides not present in the KIV2 region [41].

Overall, despite attempts to integrate commercial assays, a single conversion factor between mg/dL of lipoprotein mass and nmol/L of apo(a) is not feasible and should be avoided in clinical practice [42]. Indeed, measurements of the Lp(a) molar concentration are the gold-standard for clinical assessment of CV risk [11].

Concerning apo(a) isoform size, three methods are available to evaluate KIV repeats on a DNA level. Pulsed field gel electrophoresis/southern blotting of genomic DNA and also fiber-fluorescence in situ hybridization [43] permits the evaluation of KIV2 copies in separated alleles. In contrast, quantitative polymerase chain reaction evaluates the sum of KIV2 copies present in investigated genomes [44].

Fasting is not required for Lp(a) testing [45] and its levels are stable over long periods of time [46,47]. However, serial measurements from placebo-treated subjects in clinical trials have shown that intraindividual temporal variability could be up to 20%. 

Current guidelines [48,49] suggest a cut-off of 50 mg/dL (125 nmol/L) to define hyperlipoprotein(a), though a gradual increase in risk has been found from values over 30 mg/dL [7,50]. In fact, in a large meta-analysis involving 126′634 participants and more than one million person-years of follow-up, an increased CV risk with levels of Lp(a) > 24 mg/dl was found [51] with results similar to the Mendelian Randomization (MR) studies [7,10]. Lp(a) concentrations significantly differ among populations, with Caucasians having the lowest levels and black Americans having the highest one [52,53]. This opens to different cut-off levels depending on the ethnicity [7]: indeed, the Multi-Ethnic Study of Atherosclerosis found that the 50 mg/dl cut-off can be used for white Americans, while a threshold of 30 mg/dl is more appropriate for black Americans [46,49].

The European Society of Cardiology (ESC) very recently published a consensus statement on Lp(a) [54]. As also previously stated in the 2019 ESC guidelines on dyslipidaemias [55], they recommend that Lp(a) level should be measured at least once in the lifetime in all individuals, in order to identify the ones with a very high level (≥180 mg/dL or ≥430 nmol/L). In these subjects the risk of CV diseases can be compared to that of subjects affected by heterozygous familiar hypercholesterolemia [54].

## 5. Lipoprotein(a) and Cardiovascular Disease: Evidences and Possible Mechanisms

The relationship between Lp(a) and CV events has been well established in epidemiological and GWAS studies [9,10] as well as in meta-analyses [4,5] and it is considered the strongest single genetic risk factor known for CV diseases [36]. This fact makes it an excellent candidate for MR studies [6,7,8].

Many CV events have been linked to Lp(a). Starting with acute coronary syndrome [46,56], a recent MR study found that 2-fold higher level of genetically determined Lp(a) is associated with a 22% increase of myocardial infarction risk [39].

Regarding cerebrovascular events, Sun et al. found a significant relationship not only with ischemic stroke (lacunar and atherothrombotic) but also with hemorrhagic stroke [57]. Even after the correction for the classic CV risk factors, Lp(a) caused a 1.97-fold increase in the risk of overall stroke [57]. A MR study that analyze 2 combined Danish cohorts [6] found a role for Lp(a) as a stroke determinant. Additionally, its role is further supported by genetic studies in which 4 apo(a) SNPs, associated with low Lp(a) levels, were associated with a 13% lower risk of stroke and a 30% lower risk of acute coronary syndrome [43].

Furthermore, Lp(a) levels are associated with Peripheral Arterial Disease (PAD) [58]. In particular, two recent studies have shown that Lp(a) levels correlate with new peripheral lesion, repeated peripheral artery revascularization and major amputation in a population with PAD [59,60].

The role of Lp(a) in CV diseases is not limited to atherosclerotic lesions, but also to cardiac arrhythmia and valvular diseases. Regarding arrhythmias, most of the studies focused on atrial fibrillation with conflicting results.

No correlation was found between genetic variants of Lp(a) genes and incidence of atrial fibrillation in one study [61] while others found an inverse relationship (i.e., lower incidence of atrial fibrillation with higher Lp(a) levels) [62,63,64]. In one of these studies the inverse relationship was confirmed only in women without chronic coronary syndrome [47].

Conversely, a more recent study [65] based on UK Biobank data (N = 435,579) found a positive correlation, with an increased incidence of atrial fibrillation by 3% for each 50 nmol/L (23 mg/dL) increase in Lp(a). So, more studies are needed on this topic in order to better understand the direction of the relationship (positive vs negative) and the possible mechanisms involved.

The underlying mechanisms through which Lp(a) can determine vascular damage are multiple and can be synthesized in three categories: pro-inflammatory, pro-atherogenic and pro-thrombotic (Figure 2).

Lp(a) is more atherogenic than LDL because of apo(a) presence. Its lysine-binding sites bind tightly to the exposed surfaces on endothelium, infiltrate the vessel wall and accumulate into subintimal spaces [66]. This binding also up-regulates adhesion molecules and stimulates the proliferation of smooth muscle cells [67]. Similarly to LDL, when inside the plaque, Lp(a) can oxidize creating a highly pro-inflammatory and immunogenic oxidized complex [68]. As already mentioned Lp(a) is the most important plasma carrier of oxidized phospholipids that contributes to the high levels of plaque inflammation, being able to up-regulate inflammatory genes and cytokines release [69]. In addition, in presence of high Lp(a) levels, monocytes have a greater production of pro-inflammatory cytokines and present an enhanced penetration capacity through the arterial wall [70]. Monocyte activation results in foam cells formation, cellular apoptosis, acceleration and enlargement of the necrotic core [71]. All these factors can be summarized in the result of some studies that found an increased accumulation of 18-fluorodeoxyglucose in carotid arteries and aorta of individuals with high Lp(a) levels [72].

Finally, Lp(a) has a pro-thrombotic potential, reducing plasminogen activation and fibrine degradation while increasing Plasminogen Activator Inhibitor-1 expression on the endothelial cells and the activity of the tissue factor pathway inhibitor. All these mechanisms results in a more intense platelet activation and thrombus formation [53].

## 6. Lipoprotein(a) and Aortic Valve Stenosis

The pathological link between Lp(a) and Aortic Valve Stenosis (AVS) has been demonstrated in many studies. Among them, a 2013 GWAS for the first time suggested that Lp(a) is strongly associated with aortic valve calcification and AVS [73]. This was also confirmed in prospective studies in which the progression of AVS was found to be related to Lp(a) values [74]. In particular, elevated Lp(a) level is associated with faster progression of the narrowing and increased need of aortic valve replacement [75].

However, a recently published paper found Lp(a) to be implicated in the onset but not in the progression of aortic valve calcification [76]. The process leading to AVS is long and starts with cusps calcification. This is similar to the atherosclerotic process with which it shares risk factors (age, obesity, smoking, hypertension, dyslipidemia) and initial steps. In fact, the calcification process begins with endothelial damage of the cusps with subsequent lipid infiltration and oxidation, triggering an inflammatory process that involves macrophages, T-lymphocytes, and mast cells [77]. In the following step (propagation), calcification and fibrosis lead to the activation of remodeling process of the valve that is similar to those involving the bone [78].

The link between Lp(a) and AVS is mainly mediated by the Lp(a) capacity to bind the endothelial surface and infiltrate the inner layers of the aortic valve [79]. Here, the oxidated phospholipids are converted by the enzyme Lp-phospholipase-2 into lysophosphatidylcholine which promotes valve mineralization. This molecule is further converted into lysophosphatidic acid through the enzyme autotaxin present on Lp(a). It further stimulates the valvular interstitial cells to produce osteoblastic transcription factors, runt-related transcription factor 2 and bone morphogenetic protein 2 [80]. All these steps determine the differentiation of valvular interstitial cells to osteoblasts-like cells initiating the process of valvular calcification and AVS development. Bone morphogenetic protein 2 increases alkaline phosphatase, which provides inorganic phosphate that further powers mineralization [81]. Furthermore, as already mentioned, oxidated phospholipids promote inflammatory response and macrophages, T-lymphocytes and mast cells produce widespread microcalcifications within the endothelium [82].

Due to the complexity of this pathogenic process, many efforts have been made to find a pharmacological target in order to prevent or arrest the aortic degeneration, unfortunately with no significant results [83]. Angiotensin-converting-enzyme inhibitors, angiotensin receptor blockers, eplerenone, nitrates, statins, denosumab and alendronic acid failed to modify the disease progression [75,84].

Instead, a post-hoc analysis of the FOURIER trial with evolocumab highlighted the Lp(a) reduction as a therapeutic option for AVS [85]. In this analysis, patients treated with evolocumab had a numerically lower incidence of new or worsening AVS or valve replacement when compared with placebo (0.27 vs. 0.41%) with a significant HR of 0.48 (95% CI, 0.25–0.93). Moreover, trials with niacin (vs placebo; EAVaLL trial; NCT02109614) and PCSK9 inhibitors (NCT03051360) are actually ongoing. Probably, in the near future some of the new specific approaches for reducing Lp(a) (see the next paragraph) will also be tested in specific studies on AVS, in order to understand if Lp(a) reduction is also able to prevent or slow the progression of aortic calcification.

Mitral valve shares with aortic valve a similar pathological process, that leads to stenosis; unfortunately few data are available on its relationship with Lp(a). While one study found an association with mitral stenosis [86], another one found it only for mitral calcification [87]. Interestingly, Lp(a) was associated with mitral valve calcification onset but not with its progression over time [88], as it was also for aortic valve calcification. It is possible that the different prevalence of mitral stenosis, as well as differences in flow condition/valve anatomy and the evaluation modality (anatomical with computer tomography [87,88] or flow-related with echocardiography [86]) could explain conflicting results.

## 7. Lipoprotein(a): Treatment

Despite the well established association between high levels of Lp(a) and CV events, no specific recommendations on drug therapy to lower Lp(a) have been given in expert statements and guidelines on the management of dyslipidaemias [53,89,90]. The latest consensus statement from ESC [54] focuses on intensifying risk factor management in those subjects at higher risk in order to reduce the global CV risk.

While lowering Lp(a) may lead to CV risk reduction, the currently available therapies are not specific (they mainly act on LDL or other lipoproteins) and/or determine only a small decrease in plasma values of Lp(a). It has been estimated that a major reduction of Lp(a) is needed to exert the same effects obtained through LDL reduction. Indeed, a 65.7 mg/dL Lp(a) reduction leads to the same effects of a 38.6 mg/dL LDL reduction [91]. Furthermore, Lp(a) reduction will affect the residual risk, i.e., the percentage of risk for future events after optimal medical therapies.

Although the recurrence of CV events (also called extreme CV risk [92]) affects at least 10% of patients with a previous acute coronary syndrome, it is possible that the effects determined by a weak Lp(a) reduction are over-shaded by the intense control of other CV risk factors and the relative benefit in terms of risk reduction. However, it is expected that a strong decrease of Lp(a) will lead to positive effect on CV events; this is currently tested in specific trials.

Although they are in general beneficial, lifestyle changes, such as low fat diet and moderate-to-vigorous daily physical activity, have no significant effect on Lp(a) [93,94].

Regarding drugs, both commercially available and experimental therapies have been summarized in Table 1.

The effect of statins on Lp(a) is controversial. Different studies showed an increase of Lp(a) levels in subjects receiving statin therapy, although this association seems to be present only in patients with a small apo(a) phenotype. Indeed, after 2 months of statins treatment patients with a small molecular mass apo(a) showed an increase in Lp(a) levels (from 66.5 to 97.4 mg/dL, *p* = 0.026) while this is not the case for subjects with a large molecular mass [95]. In these patients, only a slight and non significant increase was found (from 20.5 to 23.3 mg/dL, *p* = 0.09). Nevertheless, statins are indicated in patients with high Lp(a) in order to reduce the overall risk with an intense LDL lowering [53].

Niacin determines a 20% decrease in Lp(a), a decrease in LDL level of 10 mg/dL and in triglycerides level of 33 mg/dL and an increase in HDL level of 6 mg/dL [96,97]. However, this reduction does not cause a decrease in CV events, as determined in two clinical trials (AIM-HIGH and HPS2-THRIVE) [98,99]. Furthermore, the combined therapy with niacin and statins increases the risk of serious adverse effects (diabetes onset, musculoskeletal system symptoms, infection and bleeding) [97].

Lp(a) levels reduction with anti-PCSK9 antibodies was analyzed in two post-hoc papers of the registrative trial of evolocumab (FOURIER) [100] and alirocumab (ODYSSEY OUTCOMES) [101]. In particular, evolocumab determines a 29.5% [93] Lp(a) reduction while alirocumab decreased its levels by 23.5% [90]. Both of them showed that Lp(a) levels lowering was independently associated with a reduction in CV events [97,99], bringing some hope that these drugs may become useful in patients with high levels of Lp(a). This reduction is probably determined by the fact that LDL receptor is highly exposed on hepatocyte membrane with very low levels of LDL in circulation. This probably determined a higher Lp(a) molecules uptake as well: in fact, production does not seem to change [19]. Also inclisiran, a small-interfering RNA (siRNA) inhibiting PCSK9, has been evaluated in terms of Lp(a) reduction resulting in a 25.6% lowering [102].

Similarly to statins, all PCSK9 inhibitors are actually (or in the near future for inclisiran) used to target LDL cholesterol with modest effects on Lp(a). Reducing the risk of MACE by targeting Lp(a) may require higher reductions with more powerful and specific therapies.

Mipomersen is a 2-O-methoxy-ethyl-modified antisense oligonucleotide—ASO—targeted to apo(B) mRNA that decreases apoB synthesis by inhibition of messenger ribonucleic acid translation. It is able to reduce LDL-C and Lp(a) both in the range of 21–50%, but its side-effect profile is such that its use is restricted to subjects with homozygous FH, and it is not appropriate for routine Lp(a) lowering [103].

Lipoprotein apheresis is actually the only therapy (non pharmacological) that determines an important decrease (60–70% acutely after each therapeutic session) in Lp(a) [104]. It exchanges 4 to 6 L of plasma in 2 to 4 h weekly or biweekly [105] and leads to a similar decrease in LDL cholesterol. Despite the acute important decrease in Lp(a), regular sessions of apheresis results into a lower mean Lp(a) reduction (between 25% and 40%) [104]. There are different systems used for apheresis, the most common ones share the specific adsorption of apoB, constitutive of VLDL, LDL and Lp(a) [106]. Although it is mentioned in lipid guidelines, they differ significantly in defining which patients need to be treated with apheresis [107]. It can be used only for LDL cholesterol reduction in patients with homozygous familial hypercholesterolemia refractory to lipid-lowering drugs or with an inadequate response [106,107,108,109]. In these patients, it needs to be started as soon as possible, preferably in early childhood. The situation is less clear for patients with heterozygous familial hypercholesterolemia or hyperlipoprotein(a). Despite the procedure is associated with minimal side effects, it is very costly and time spending; therefore, it should be reserved for patients with very high Lp(a) levels, and progressive and/or severe CV disease.

The most interesting approach for the future comes from new specific drugs. In particular an ASO and three siRNA are actually under evaluation [110]. Both of them have a subcutaneous administration. IONIS-APO(a)Rx is a second generation 2-O-(2-methoxyethyl)-modified ASO created to reduce the synthesis of apo(a) with a weekly administration [109]. Pelacarsen (TQJ230 or AKCEA-APO(a)-LRx) is an evolution of IONIS-APO(a)Rx with a N-acetyl-galactosamine complex that allows specific uptake via the asialoglycoprotein receptor in hepatocytes, determining a 30-fold higher potency and can be administered monthly.

The two dose finding phase 1/2 trials were published together with the finding of a reduction up to 72% with IONIS-APO(a)Rx and up to 92% with Pelacarsen, using the highest dose [111]. Pelacarsen continued its step with a phase 2b multicenter international randomized, double-blind, placebo-controlled, dose-ranging trial on 286 patients with established CVD and baseline Lp(a) > 60 mg/dL. The highest dose was associated to a 80% Lp(a) reduction and was well tolerated, the most common adverse effects being injection site reactions (7%) and erythema (26%) [112]. Interestingly, no differences in the effects on Lp(a) were found with respect to the different apo(a) isoform [113]. A multicenter trial assessing the impact of Lp(a) lowering with Pelacarsen on major cardiovascular events in patients with established CV disease (HORIZON; NCT04023552) is actually ongoing. The enrollment of 8323 patients has been completed on March 2022 and the study is planned to end in May 2025. Inclusion criteria were the presence of a previous CV event (acute coronary syndrome, ischemic stroke or peripheral lower limbs artery disease) in the last 10 years, Lp(a) ≥ 70 mg/dL and optimized CV risk factor therapy (for instance, LDL < 55 mg/dL); the trial will provide evidence on whether the addition of Lp(a) lowering drug to intensive modern secondary prevention treatment can lead to a further decrease of CV events. Furthermore, other 2 trials are ongoing with this Pelacarsen: one that tests the reduction in lipoprotein(a) apheresis after the begin of the therapy (NCT05305664) and a phase 1 trial in patients with mild hepatic impairment (NCT05026996). Regarding siRNA three drugs of this class are under evaluation. The first one is Olpasiran (AMG890) which has shown a reduction in Lp(a) between 80% and 94% at day 113 in patients with Lp(a) levels between 70 nmol/L and 199 nmol/L after a single dose injection, whereas the compound was less effective in individuals with Lp(a) levels ≥ 200 nmol/L [114]. In the phase 2 study (OCEAN(a)-DOSE; NCT04270760) 4 different doses have been tested with subcutaneous administration every 3 months (10, 75 and 225 mg) or every 6 months (225 mg) on 281 patients [115]. The trial has recently been concluded with official results yet to be published. However, recent congresses’ report showed a reduction of 94% with the 75 mg every 3 months dose. The reduction reaches 97 and 98% with the 225 mg every 3 or 6 months dose without safety concerns. The phase 3 trial was recently (October 2022) registered on clinicaltrial.gov (NCT05581303) with the estimated starting date in December 2022. Furthermore, the producer company has already started 3 other studies: 2 phase one trial to test the drugs in the context of renal (NCT05489614) and hepatic (NCT05481411) impairment, and 1 epidemiological study (NCT05378529) in order to assess the prevalence of elevated Lp(a) values in 20,000 patients that already suffered an acute coronary syndrome.

Another siRNA (SLN360) has recently completed the phase 1 trial with a single ascending dose [116]. With the highest dose (600 mg) a 98% reduction in Lp(a) was found to persist at least for 150 days. No significant safety concerns were founded in the study. On September 2022 the phase 2 trial was registered on clinicaltrial.gov (NCT05537571) with the estimated starting date in December 2022.

Finally, LY3819469 have an ongoing phase 1 clinical trial (NCT04914546). The study is finished but results have not been already presented. However, a phase 2 study is now registered on clinicaltrial.gov (NCT05565742, registration October 2022).

## 8. Conclusions

Lp(a) is a strong risk factor for CV events, but also for AVS onset and progression. Many studies and consensus papers on methods and clinical indication for Lp(a) measurements have been published, as well as on the cut-off levels that should be used. Some of the drugs acting on LDL cholesterol are also able to reduce Lp(a) but in a modest way, probably insufficient to determine a significant reduction of CV events. However, one ASO and 2 siRNA already demonstrated a strong reduction (from 80 to 90%) with a favorable safety profile. One phase 3 trial is ongoing with Pelacarsen and another one is planned to start with Olpasiran. The results of these studies will provide a definitive answer on whether it is indicated to reduce Lp(a) in patients with previous CV events.

## Figures and Tables

**Figure 1 ijms-24-00170-f001:**
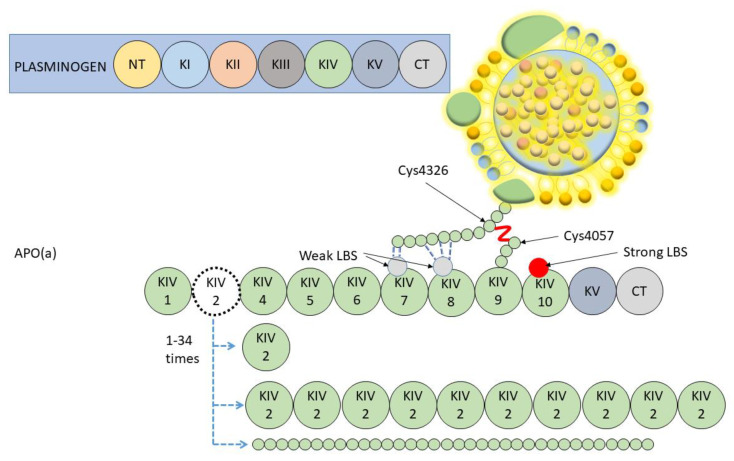
Lp(a) and plasminogen structural homology. K = Kringles (numbered from I to V); CT = C-Terminal; NT = N- Terminal; LBS = Lysine Binding Site.

**Figure 2 ijms-24-00170-f002:**
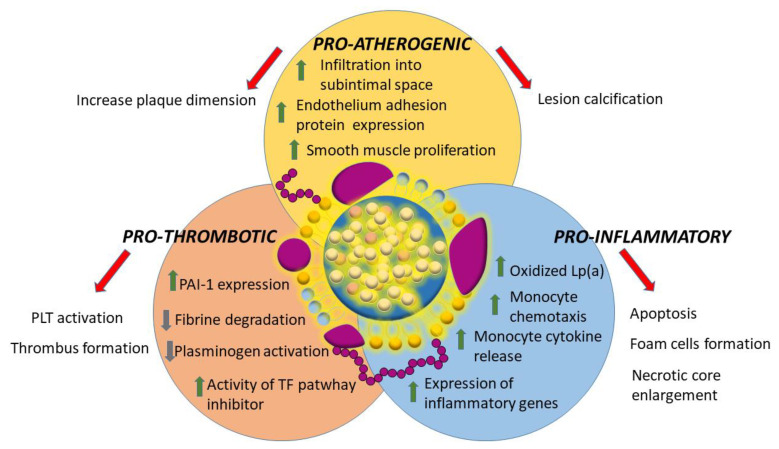
Summary of the three main mechanisms of Lp(a) mediated cardiovascular damage. PLT = Platelets; PAI-1 = Plasminogen activator inhibitor type 1; TF = Tissue Factor.

**Table 1 ijms-24-00170-t001:** Non-specific and specific therapies for Lipoprotein(a) reduction. HMG-CoA = HydroxyMethylGlutaryl CoA; PCSK9 = Proprotein Convertase Subtilisin/Kexin-type-9; siRNA = small interfering RiboNucleic Acid; ApoB = Apolipoprotein B; ASO = antisense oligonucleotide.

**Non-Specific Therapies**				
**Name**	**%Lp(a) Reduction**	**Mechanism of Action**	**Administration**	**Experimental Phase**
Statins	controversial	HMG-CoA reductase inhibitor	Oral	No
Niacin	20	Triglycerides production inhibitor	Oral	No
Alirocumab	23.5	Anti-PCSK9 antibodies	Subcutaneous (every 2 weeks)	No
Evolocumab	29.5	Anti-PCSK9 antibodies	Subcutaneous (every 2 weeks)	No
Inclisiran	25.6	siRNA inhibiting PCSK9	Subcutaneous (every 6 months)	No
Mipomersen	21–50	Decreases apoB synthesis	Subcutaneous (weekly)	No
**Specific Therapies (All Experimental)**				
**Name**	**%Lp(a) Reduction**	**Mechanism of Action**	**Administration**	**Experimental Phase**
Pelacarsen	80	ASO inhibiting apo(a)	Subcutaneous (every 1 month)	Phase 3 ongoing (patients enrollment complete), result expected for mid 2025
Olpasiran	94–98	siRNA inhibiting apo(a)	Subcutaneous (every 3 months)	Phase 2 finished (results not yet published), Phase 3 planned to start in December 2022
SLN360	98	siRNA inhibiting apo(a)	Subcutaneous (probably every 6 months)	Phase 1 finished, Phase 2 planned to start in December 2022
LY3819469	unknown	siRNA inhibiting apo(a)	Subcutaneous (timing unknown)	Phase 1 finished but results not yet presented, Phase 2 registered on clinicaltrial.gov

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
