# Peer review of "Lipoprotein(a): Cardiovascular Disease, Aortic Stenosis and New Therapeutic Option"

_ijms, 2022, doi:10.3390/ijms24010170_

Round 1

Reviewer 1 Report

The manuscript aims to review the role of Lipoprotein (a) in cardiovascular disease states and summarize how its plasma level correlates to increased risk of CVD and AVS.  They also discuss current therapeutic approaches to lowering LDL and Lp(a), as well as emerging new therapies using siRNA. A nice table was added summarizing current and experimental therapies in clinical trials.  The review appears to be comprehensive but needs a lot of editing (syntax/grammar), which decreases reader enthusiasm. 

Comments:

1.        The manuscript needs to be professionally edited.  There are many non-scientific (NS) words, missing punctuations, misspelled words, and run-on sentences.  Listed below are a few examples:

The word “also” was used very often without necessity.

Line 12:  In addition to the classic Cardiovascular (CV) risk factors, in the latest years also (NS) other new markers able to increase the risk of CV disease have been identified….. new markers capable of increasing CVD risk have been recently identified.

Line 21: going through the finding that confirms (NS) its association… examine its association

Line 28: worldwide (redundant)

Line 34: in the latest years also (NS) other new… recently other new

Line 82:  The main determinant of plasma levels is genetics and depends on the codominant expression of 2 apo(a) alleles (located at positions 26 and 27 on the long arm of chromosome 6 - 6q26-27) being the final circulating levels determined by the sum of contribution by each allele. Run-on sentence.  The main determinant of plasma levels is genetics and depends on the codominant expression of 2 apo(a) alleles (located at positions 26 and 27 on the long arm of chromosome 6 - 6q26-27).  Final circulating Lp(a) levels are determined by the sum contributed by each allele.

Line 90: as also shown by the fact that (NS) in genetically unrelated individuals carrying the same isoform size combination the plasma concentration can vary by up to 200-fold… as shown in genetically unrelated individuals carrying the same isoform size combination where plasma concentration can vary by up to 200-fold.

Line 134: most important receptor (singular)

Line 142:  peculiar (NS) perhaps use the word unique; covalent bond (not bound)

Line 172: harmonize (NS) perhaps integrate

Line 233: Al already….. As already

Line 246: hesitate?.... results

Line 250: evidences…. studies

2.       Line 75:  the LBS involved in the association between apo(a) and apoB is a possible target for Lp(a) reduction using small molecules specifically targeted.  

Please clarify.  Are small molecules (which ones?) being used to target the LBS binding site?

3.       Line 78: Is there a specific phospholipid being oxidized or just phospholipids in general?

4.       There are several instances in the manuscript where the references used refers to another review paper instead of the original manuscript and with very similar language.

Line 123-125: The site of assembly is still controversial, in vitro studies have shown that the binding between apo(a) and apoB may occur either at the surface of the hepatocytes or in the space of Disse [21].

Line 228-230: In fact, its lysine-binding sites bind tightly to the exposed surfaces on endothelium, infiltrate the vessel wall and accumulate into subintimal spaces more intensively than LDL [63].  **This reference is about the correlation of PE and Lp(a) for which none was found.  The information for the sentence above was taken from the introduction of this review.  This is not the correct way to refence material.

5.       Line 133:  How can LDLR be the most important receptor for Lp(a) clearance even though it has a small affinity to LDLR?  Seems contradictory.  Are there other receptors for which it has higher affinity?

6.       Figure 2: Font should be increase inside the circles; Why doesn’t the pro-atherogenic not have arrow indicating downstream events like the other two?

Reviewer 2 Report

It's a very high-quality full-fledged literary review. I would like to especially note the illustrative part, as well as the summary table on drugs, which makes the article more interesting and understandable for the reader.

The only minor remark may be the absence of reflection of Lp(a) associations and other cardiovascular diseases (for example, mitral stenosis, cardiac arrhythmias), as well as atherosclerosis of other localizations (PAD, etc.). However, this remains at the discretion of the authors.
